# Novel Functions of the Fatty Acid and Retinol Binding Protein (FAR) Gene Family Revealed by Fungus-Mediated RNAi in the Parasitic Nematode, *Aphelenchoides besseyi*

**DOI:** 10.3390/ijms221810057

**Published:** 2021-09-17

**Authors:** Shanwen Ding, Chunling Xu, Chun Chen, Junyi Li, Jiafeng Wang, Hui Xie

**Affiliations:** Laboratory of Plant Nematology and Research Center of Nematodes of Plant Quarantine, Department of Plant Pathology, College of Plant Protection, South China Agricultural University, Guangzhou 510642, China; dingshanwen@foxmail.com (S.D.); xuchunling@scau.edu.cn (C.X.); chchun@scau.edu.cn (C.C.); 13427583582@163.com (J.L.); jfwang@scau.edu.cn (J.W.)

**Keywords:** *Aphelenchoides besseyi*, FAR gene, fungus-mediated RNAi, *Botrytis cinerea*

## Abstract

RNA interference (RNAi) is a powerful tool for the analysis of gene function in nematodes. Fatty acid and retinol binding protein (FAR) is a protein that only exists in nematodes and plays an important role in their life activities. The rice white-tip nematode (RWTN), *Aphelenchoides besseyi*, is a migratory endoparasitic plant nematode that causes serious damage in agricultural production. In this study, the expression levels of eight RWTN genes were effectively decreased when RWTN was fed *Ab-far-n* (*n*: 1–8) hairpin RNA transgenic *Botrytis cinerea* (ARTBn). These functions of the *far* gene family were identified to be consistent and diverse through phenotypic changes after any gene was silenced. Such consistency indicates that the body lengths of the females were significantly shortened after silencing any of the eight *Ab*-*far* genes. The diversities were mainly manifested as follows: (1) Reproduction of nematodes was clearly inhibited after *Ab-far-1* to *Ab-far-4* were silenced. In addition, silencing *Ab-far-2* could inhibit the pathogenicity of nematodes to *Arabidopsis*; (2) gonad length of female nematodes was significantly shortened after *Ab-far-2* and *Ab-far-4* were silenced; (3) proportion of male nematodes significantly increased in the adult population after *Ab-far-1*, *Ab-far-3*, and *Ab-far-5* were silenced, whereas the proportion of adult nematodes significantly decreased in the nematode population after *Ab-far-4* were silenced. (4) Fat storage of nematodes significantly decreased after *Ab-far-3*, *Ab-far-4,* and *Ab-far-7* were silenced. To our knowledge, this is the first study to demonstrate that *Ab-far* genes affect sex formation and lipid metabolism in nematodes, which provides valuable data for further study and control of RWTNs.

## 1. Introduction

The rice white-tip nematode (RWTN), *Aphelenchoides besseyi*, is a migratory and seed-borne plant parasitic nematode (PPN) that infects the aboveground parts of plants and causes rice white-tip disease. Its occurrence has been recorded in most rice-growing areas worldwide. RWTN and other plant nematodes cause an economic loss of approximately USD 1.6 billion for rice every year [1,2]. To prevent the spread of RWTN, some countries have classified it as a quarantine plant pest [3]. Concerns over environmental safety have caused the deregistration of some frontline nematicides, leading to serious shortfalls in the efficacy of current nematode management strategies [4]. Therefore, it is necessary to further study and reveal the parasitic process of RWTN to develop safer and more effective strategies to control RWTN.

Nematodes can secrete lipid-binding proteins that play potential roles in nutrient absorption, host tissue regulation, and host defense response inhibition [5,6,7]. One of these proteins is the fatty acid and retinol binding protein (FAR), which can be secreted through the body wall to help nematodes obtain fatty acids and retinols from the environment or hosts. Fatty acids and retinols are necessary for nematode life activities; however, the nematode itself cannot synthesize them. Fatty acids and retinols play important roles in physiological processes, such as gene activation, cell signaling pathways, tissue differentiation, and repair, and may inhibit or interfere with the host’s defensive immune response [8,9]. As a unique protein in nematodes, more than 70 FAR sequences and functions have been identified in different species of nematodes [6,7,8,9,10,11,12,13,14,15,16,17]. The function of the FAR protein gene (*far*) in five species of plant nematodes has been studied and verified by RNA interference (RNAi) [8,12,15,18,19,20]. Among them, RWTN *Ab-far-1* [12], *Radopholus similis Rs-far-1* [18], *Heterodera avenae Ha-far-1* [19], and *Pratylenchus penetrans*
*Pp-far-1* [20] were targeted via RNAi. Altogether, the expression of *far-1* and the reproduction of nematodes were significantly decreased, the pathogenicity of *R. similis* against *Anthurium* was significantly reduced, *Rs-far-1* was demonstrated to be involved in the regulation of allene oxide synthase (AOS) expression in *Arabidopsis*, and *Mj-far-1* hpRNA transgenic tomato could reduce the infection of *Meloidogyne javanicae*, whereas Mj-FAR-1 protein might regulate lipid signaling and induce the host’s sensitivity to the nematode [8,15]. As the *far* genes occur exclusively in nematodes, and play important roles in growth and development, parasitic and pathogenic process, and other life activities of nematodes, using *far* genes as targets to control PPNs will not harm either the environment or other organisms. Therefore, studies on the *far* gene family have become a hotspot in research.

At present, eight genes (*Ab-far-1*~*Ab-far-8*, GenBank accession number: JQ686690, KT387726–KT387732) have been identified and cloned from RWTN [12,17]. The eight *Ab-far* genes were all expressed in different developmental stages of nematodes, and their patterns of expression in RWTN were found to differ. *Ab-far-2*, *Ab-far-6*, *Ab-far-7*, and *Ab-far-8* showed specific localization, *Ab-far-2* and *Ab-far-7* were expressed in the genital system, and *Ab-far-6* and *Ab-far-8* were expressed in the esophageal glands. *Ab-far-1*, *Ab-far-3*, *Ab-far-4*, and *Ab-far-5* were expressed in multiple locations of RWTN, including the digestive system, reproductive system, and body wall. In addition, *Ab-far-3* and *Ab-far-5* were found to be expressed in the nerve ring of RWTN. These findings suggest that FAR proteins of RWTN may be involved in many biological activities and show diversity in strategies for obtaining fatty acids and retinol with FAR proteins for lipogenesis in nematodes [12,17]. Therefore, it is of great significance to further study the eight *Ab-far* genes of RWTN.

PPNs are generally obligated plant parasites that lack specific mutants and transformation systems. As a result, there are many limitations in the forward genetic analysis of PPNs. RNAi is an important method of reverse genetics and has become the most effective tool for studying the function of the PPN gene. In our previous study, we established a fungal-mediated RNAi method for plant nematodes by feeding these organisms the target gene hairpin RNA (hpRNA) transgenic *Botrytis cinerea* [21]. Compared with the RNAi effect produced by in vitro RNAi, this method enables persistence and heritability. Compared with *in planta* RNAi, this method has the advantage of simple operation, high conversion rate, good reproducibility, and a shorter period [21]. Therefore, the functions and specific roles of the eight *Ab-far* genes were further studied using fungi-mediated RNAi. It is of great significance and value to reveal the development process and pathogenesis of nematodes, and to establish a new method of controlling nematodes by using *far* genes.

## 2. Results

### 2.1. Production and Identification of Transgenic B. cinerea

The constructed RNAi vectors, pBS-Ab-far-*n* (*n* = 2, 3, 5, 6, 7, and 8) expressing *Ab-far-n* hpRNA (Figure 1A), were transferred into *B. cinerea* mediated by *A. tumefaciens*. These ARTB*n*s were generated on PDA solid medium supplemented with 100 μg/mL hygromycin and identified by PCR, sequencing, and Southern blot. The results showed that the sizes of fragments obtained by PCR were consistent with the predicted sizes. The fragments ARTB*2* No. 2, ARTB*3* No. 1, ARTB*5* No. 2, ARTB*6* No. 3, ARTB*7* No. 3, and ARTB*8* No. 3 had a single-copy insertion in their respective genomes. No hybridization signal was detected in the WTB genomic DNA (Table 1; Figure 1B). These results indicate that hpRNAs were successfully inserted into *B. cinerea* genomic DNA. Transgene-induced RNAi in single-copy lines is reportedly more effective than in other lines [22,23] and is convenient for further analysis of the next generation. The single-copy ARTB*1* and ARTB*4* preserved in the laboratory were also tested by PCR and verified by sequencing. These single-copy ARTBns were selected and labeled ARTB*1*-ARTB*8* for use in subsequent experiments in this study.

### 2.2. Ab-far-n hpRNA Had No Effect on Either the Growth, Carbohydrate, or Protein Contents of B. cinerea

Carbohydrates and proteins are essential nutrients for animals to maintain their normal activities. *Caenorhabditis elegans* feeding on *Escherichia coli* with different nutrient contents could affect many life characteristics [24,25]. Therefore, to ensure that the phenotypic changes of RWTN were caused by silencing of the *Ab*-*far* genes, the growth, carbohydrate, and protein contents of *B. cinerea* were tested and analyzed. The results showed no significant differences in these indicators between ARTB*n* and WTB (Figure 2). When each strain of *B. cinerea* grew to the sixth day, each could grow to 100% confluency in a 60 mm Petri dish and start sporulation. Further, their carbohydrate and protein contents were approximately 30 mg/g and 900 ug/g, respectively.

### 2.3. Ab-far-n Silencing Efficiency and Reproduction of RWTN Fed ARTBn

RT-qPCR was used to detect the silencing efficiency of the corresponding target gene of RWTN from different treatments. The results showed that the expression levels of *Ab-far-n* of RWTN fed T*n* were significantly decreased compared with those of RWTN fed WTB or GTB (*p* < 0.05 for both), whereas the latter did not significantly differ from each other (Figure 3A, Table 2). These results indicate that the target gene expression levels of RWTNs were efficiently silenced when the nematodes were fed the transgenic *B. cinerea*. The reproduction of RWTNs from different treatments (Table 2) was arranged in descending order as follows: T*8–7*, GT, WT, T*5–6*, T*3–4*, T*1*, and T*2*. The reproduction of RWTN from T*1* to T*4* treatments was significantly decreased by 44.93%, 65.83%, 32.00%, and 33.76%, respectively, compared with the WT treatment (*p* < 0.05 for each), and significantly decreased compared with the GT treatment (*p* < 0.05). Except for the T*1* to *T4* treatments, other treatments showed no significant difference relative to WT and GT, or between the WT and GT groups. In conclusion, silencing *Ab-far-1* to *Ab-far-4* significantly inhibited the reproductive ability of RWTN, and silencing *Ab-far-2* had the strongest inhibitory effect.

### 2.4. The Proportion of Females, Males, Juveniles and Adults of Nematodes after Silencing Ab-far-n of RWTN

Our statistical results on the proportion of females, males, juveniles, and adults from each treatment (Table 2) showed that the proportion of males was significantly increased from 4.47% in the WT treatment to 13.24%, 12.30%, and 13.16% in the T*1*, T*3*, and T*5* treatments (*p* < 0.05 for each), respectively, whereas the proportion of females was decreased from 53.75% in the WT treatment to 46.08%, 40.74% and 40.42% in the T*1*, T*3*, and T*5* treatments (*p* < 0.05 for each), respectively. The proportion of juveniles significantly increased from 41.78% in the WT treatment to 49.57% in the T*4* treatments (*p* < 0.05), whereas the proportion of adults was significantly decreased from 58.22% to 50.43% (*p* < 0.05). There was no significant difference between WT and GT treatments, indicating that the ratio of females, males, and juveniles was not affected by feeding on the exogenous gene (eGFP) hpRNA transgenic fungi (Figure 3B). These results indicate that *Ab-far-1*, *Ab-far-3*, and *Ab-far-5* might affect the sex differentiation of nematodes, and *Ab-far-4* might be involved in the development of nematodes from juveniles to adults.

### 2.5. Morphological Characteristics of Nematodes after Silencing Ab-far-n of RWTN

To determine the morphological changes in nematodes, we randomly selected 20 females and 20 males from each treatment (Table 2) for morphological measurements. Compared with the WT treatment, the body length of females in T*1*-T*8* treatments was significantly shorter (*p* < 0.05), and the gonad length of females in the T*2* and T*4* treatments was significantly shorter (*p* < 0.05). The body length and gonad length of females were not significantly different between the WT and GT treatments (Table 3). These results indicate that each *Ab-far* (*Ab-far-1*~*Ab-far-8*) might affect the growth and development of females in RWTN, and *Ab-far-2* and *Ab-far-4* might be related to the development female RWTN. There were no significant differences in any of the morphological characteristics of males among all treatments (Appendix A), indicating that silencing *Ab-far* might not affect the development of male RWTN.

### 2.6. The Pathogenicity of Nematodes to Arabidopsis Thaliana after Silencing Ab-far-n of RWTN

Previous studies have demonstrated that the reproduction and pathogenic effects of RWTN on the host were inhibited by in vitro RNAi and fungal-mediated RNAi of *Ab-far-1* and *Ab-far-4* [12,21]. It has been confirmed that RWTN infected *A. thaliana* and caused recognizable symptoms, and proposed a method to determine the pathogenicity and reproduction rates of RWTNs using *A. thaliana* as a model host plant [26]. To test the effects of the other six genes on the pathogenicity of RWTN, 100 mixed-stage RWTNs extracted from the T*2–3*, T*5–8*, WT, and GT treatments were inoculated on the leaves of *A. thaliana* for 21 days. The results showed that the symptom severity (Figure 4B,C) and reproduction (Figure 4A) in *A. thaliana* inoculated with RWTNs from the T*2* treatment were significantly lower than those of the WT and GT treatments (*p* < 0.05, respectively), whereas the other treatments showed no significant difference compared with the WT and GT treatments. Therefore, it was speculated that *Ab-far-2*, as well as *Ab-far-1* and *Ab-far-4*, played an important role in the pathogenic process of RWTN.

### 2.7. The Fat Storage of Nematodes after Silencing Ab-far-n of RWTN

The detection results of the fat storage of RWTN from each treatment (Table 3) showed that the fat storage of RWTN from the T*2*, T*5*, and T*6* were significantly decreased by 26.87%, 26.24%, and 21.98% compared with the WT treatment (*p* < 0.05 for each), respectively (Figure 5). Although the fat storage of RWTN from the T*1*, T*3–4*, T*7*, and T*8* treatments also decreased compared with the WT and GT treatments, these differences were not significant, and the difference between the WT and GT treatments was also not significant (Figure 5). Therefore, silencing *Ab-far-2*, *Ab-far-5,* and *Ab-far-6* in RWTN might also affect the fat storage and metabolic processes of the nematodes.

### 2.8. Conclusions

In this study, the functions of RWTN’s *far* gene family were analyzed using the *B. cinerea*-mediated RNAi method, and the results showed that the body length of the female nematodes was shortened after eight far genes were silenced, indicating that all eight far genes might have influenced the development of RWTN. Nevertheless, the effects of the eight genes on reproduction, pathogenicity, the ratio of females, males, juveniles and adults, and fat storage of RWTN were diverse (Table 4).

## 3. Discussion

Preliminary progress has been made in the study of the sequence characteristics, basic functional properties, and binding activity with FAR proteins [9,13,27]. However, to date, there are still no reports on their specific functions in nematodes. RWTN is one of the most harmful nematodes in agricultural production [1]. In this study, these functions of the *far* gene family were identified to be consistent and diverse through phenotypic changes after any gene was silenced.

Most of the reported expression sites of *far* mRNA were found in the body wall of the nematode [9,15,19,28]. However, the expression locations of the eight *far* genes in RWTN were diverse, indicating that their functions might also be diverse [17]. In this study, the functions of these genes in RWTN were verified using fungal-mediated RNAi. The results showed that silencing *Ab-far-1*, *Ab-far-3*, *Ab-far-4*, and *Ab-far-5* affected multiple life activities of nematodes, which further explained the phenomenon that these genes were expressed in multiple locations of RWTN, including the digestive system, reproductive system, and body wall. Although *Ab-far-2* mRNA was only located in the gonads of RWTN [17], the results of this study revealed that silencing *Ab-far-2* could lead to the inhibition of various life activities of nematodes, indicating that *Ab-far-2* may affect other life activities indirectly by participating in the reproduction process of nematodes. The phenotypic changes in RWTN after silencing *Ab-far-6*, *Ab-far-7*, and *Ab-far-8* were relatively single compared with the complexities of *Ab-far-1* to *Ab-far-5*. This result might be related to their single expression location, or that their functions were redundant in the *far* gene family. Certainly, we did not rule out the possibility that they might have other potential functions that have yet to be studied. *Ab-far-1* not only showed diversity in expression location [12] and phenotypic changes after silencing, but its expression level was markedly higher than that of the other seven genes [17]. Therefore, *Ab-far-1* is likely to efficiently participate in a variety of life activities of nematodes, which provides a basis for the *Ab-far-1* gene as a target for the control of RWTN.

The proteins secreted by the esophageal glands and body walls of parasitic nematodes are considered to be the key factors regulating nematode-host interactions and participating in the parasitism and pathogenesis of nematodes [29,30,31]. The pathogenicity and reproduction of nematodes were significantly inhibited when *Ab-far-1*, *Ab-far-2,* and *Ab-far-4* of RWTN were silenced. Among them, *Ab-far-1* and *Ab-far-4* mRNA were both expressed in the RWTN body wall. Therefore, we propose three possible reasons for the decrease in pathogenicity of nematodes after their silencing: First, due to the decline in nematode reproduction, the number of nematodes infecting the host has decreased; second, the nematode-host interaction was blocked after they were silenced. As a result, the host’s immune response could not be suppressed; third, a combination of the above two conditions. *Ab-far-2* mRNA was only located in the gonads of nematodes. Therefore, it was speculated that *Ab-far-2* indirectly affected the pathogenic process through its involvement in the reproductive process.

If a good nutrient supply was assured, most juvenile nematodes developed into adult females. When developing juveniles experienced adverse conditions, such as a high level of intraspecific competition or impaired digestion, the proportion of male nematodes was found to increase in the adult nematode population [32,33,34]. Our experimental results also support this finding. The proportion of male RWTN fed WTB and GTRB was 4.47% and 5.52%, respectively. After silencing *Ab-far-1*, *Ab-far-3*, and *Ab-far-5* genes by feeding on their hpRNA transgenic *B. cinerea*, the proportion of male RWTN increased to 13.24%, 12.30% and, 13.16%, respectively. Therefore, *Ab-far-1*, *Ab-far-3*, and *Ab-far-5* might play a highly important role in the sex differentiation of nematodes. An increase in the male-to-female ratio may be caused by the decreased ability to obtain fatty acids and retinols following a knock-down of genes, thereby leading to nutrient deficiency in nematodes, which is similar to the increase in males due to the deficiency of nutrition in the nematode environment.

Fatty acids are important substrates for the synthesis of lipids and other macromolecular structures, and for the maintenance of normal life activities in animal bodies [27,35]. An imbalance in the fatty acid composition of nematodes leads to changes in fat storage level, delayed growth, slow movement, reduced body size, no proliferation of germ cells, reduced reproductive capacity, abnormal behavior rhythm, slow signal perception, and shortened life span of nematodes [22,36,37]. Nematodes themselves cannot synthesize fatty acids but use their lipid-binding proteins, such as FAR protein, to obtain fatty acids from the host and the environment [6]. All eight FAR proteins of RWTN could bind fatty acids and retinols [12,17]. This study showed that the fat storage in RWTN was significantly decreased after either *Ab-far-2*, *Ab-far-5*, or *Ab-far-6* genes were silenced, respectively; the body length of females decreased after either *Ab-far-1*—*Ab-far-8* genes were silenced, respectively; furthermore, silencing each of the *Ab-far-1*—*Ab-far-4* genes led to a reduction in the reproduction of RWTN. These phenotypic changes in nematodes may be related to the inhibition of nematodes by obtaining fatty acids and retinol through the FAR protein. Of note, the intake of nutrients, such as carbohydrates and proteins, has a significant impact on animal metabolism and physiological functions [25].

In this study, we found that the protein and carbohydrate contents of each transgenic *B. cinerea* did not change significantly compared with that of the WTB, which ruled out the possibility that the phenotypic changes of nematodes were caused by different nutrient intakes. Silencing *Ab-far* leads to insufficient intake of fatty acids and retinols or interferes with the operation of regulatory networks in nematodes, which affects the morphology, tissues and organs, development process, gender differentiation, reproduction, and pathogenicity of nematodes. Here, the functions of the *Ab-far* genes were studied and explained by fungus-mediated RNAi, and the inhibitory effect on fat storage was found for the first time, which further provides a theoretical basis for controlling PPNs by FAR genes.

## 4. Materials and Methods

### 4.1. Biological Material and Culture

The RWTNs used in this study were collected, isolated, and identified by the Plant Nematology Laboratory, South China Agricultural University (SCAU). Nematodes were cultured on excised carrot callus in Petri dishes (diameter: 6 cm) at 25 °C in an incubator [38]. The fungus-mediated RNAi vector (pBS-1) and full-length cDNAs of *Ab-far-1* to *Ab-far-8* were preserved as plasmids. Wild-type *B. cinerea* (WTB) strain (GIM3.47), *Ab-far-1* hpRNA transgenic *B. cinerea* (ARTB*1*), *Ab-far-4* hpRNA transgenic *B. cinerea* (ARTB*4*), and eGFP hpRNA transgenic *B. cinerea* (GRTB) were preserved by the Plant Nematology Laboratory (PNL). *Arabidopsis thaliana* (Col-0) was cultivated as previously described [39].

### 4.2. Construction of RNAi Vectors

The sense and antisense RNAi fragments (FARn-S) from *Ab-far-n* (*n*: 2, 3, 5, 6, 7, 8) open reading frames (ORFs) were amplified using primers FARncm-F1/FARncm-R1 and FARncm-F2/FARncm-R2, respectively (Appendix A). The sense PCR fragments were inserted into the vector pBS-1 digested with either *Xho*I or *SnaB*I (Thermo Fisher Scientific, Waltham, MA, USA), respectively. After sequencing, the corresponding antisense PCR fragments were inserted into the *Bgl*II and *Stu*I (Thermo Fisher Scientific) sites to form the RNAi vector, pBS2-Ab-far-n, with the *Ab-far-n* hairpin structure (Figure 1A). The DNA ligases and systems used for vector construction were based on the In-Fusion^®^ HD Cloning Kit (Takara, Shiga, Japan). In the vector construction process, the PCR fragments and new vectors were sequenced to ensure the accuracy of the insertion sequences.

### 4.3. Production and Molecular Confirmation of Transgenic B. cinerea

According to the method described by Ding et al. [21], *Ab-far-n* hpRNA transgenic *B. cinerea* were obtained and cultured at 25 °C for later use in subsequent experiments. Genomic DNA of transgenic *B. cinerea* was extracted from the hygromycin-tolerant generation, as described previously [40]. *Ab-far-n* (*n:* 1–8) hpRNA transgenic *B. cinerea* (ARTB*n*) were selected and identified by PCR with the primers (Appendix A) and used as the control. The PCR fragments obtained were sequenced to ensure accuracy. For Southern blot analysis, the primers, Hph-DIG-F/Hph-DIG-R (Table 1), were designed to amplify the DIG-labeled probes. Approximately 10 μg genomic DNA from ARTB*n* was digested with EcoRI. The digested DNA was separated on a 0.8% agarose gel and then transferred to a Hybond-N+ membrane (Amersham, Little Chalfont, UK) [41,42]. Hybridization and detection were performed with a Dig High Primer DNA Labeling and Detection Starter Kit I (Roche, Basel, Switzerland) according to the manufacturer’s instructions. Equal amounts of genomic DNA from WTB were used as controls.

### 4.4. RWTNs Cultured on Transgenic B. cinerea

The ARTB*n*, WTB, and GRTB were inoculated on potato dextrose agar (PDA) plates (diameter: 60 mm) without antibiotics. The status of transgenic *B. cinerea* growing for 2, 4, and 6 d was analyzed statistically, and the carbohydrate and protein contents of transgenic *B. cinerea* growing for 6 d were detected with the total carbohydrate content detection kit (Beijing Solar Bioscience and Technology, Beijing, China) and the plant total protein extraction kit (Beijing ComWin Biotech, Beijing, China), respectively, according to the manufacturer’s instructions. After *B. cinerea* covered the PDA plates, 20 female RWTNs were inoculated onto PDA plates and cultured for 28 d at 25 °C. Thereafter, the cultivated nematodes were tested and analyzed for target gene expression and other related phenotypes of nematodes. These treatments were labeled T*1*-T*8*, GT, and WT. Each treatment was replicated five times, and each experiment was conducted twice to confirm the results and error bars.

### 4.5. Detection and Analysis of Ab-far-n Expression

Total RNA was extracted using an RNeasy Micro kit (Qiagen, Hilden, Germany), and RNA was used as a template for cDNA synthesis with a ReverTra Ace qPCR RT kit (Toyobo, Osaka, Japan). Primers qPCRn-F and qPCRn-R (Table 1) were designed to detect the expression levels of *Ab-far-n*, and 18S rRNA (AY508035) was amplified as a reference gene using the primers, 18S-F and 18S-R (Table 1). Real-time quantitative PCR (RT-qPCR) was used to analyze the expression levels of *Ab-far-n* in RWTN and performed on a CFX-96 qPCR machine (Bio-Rad, Hercules, CA, USA) with SYBR qPCR Mix (Toyobo). The results were analyzed using the 2^−^^△△Ct^ method [43]. All experiments were performed in triplicate with three biological replicates to confirm the results.

### 4.6. Detection of the Phenotypes of RWTN

The mainly detected phenotypes of RWTN that had been fed transgenic *B. cinerea* were (1) the reproduction number, (2) the ratio of females, males, and juveniles, (3) morphological characteristics, (4) pathogenicity, and (5) fat storage of nematodes. These phenotypes were derived as follows:(1)The number of nematodes fed transgenic *B. cinerea* for 28 d was determined. Each treatment was replicated five times, and each experiment was conducted twice to confirm the results.(2)The proportion of female, male, juvenile and adult RWTN fed transgenic *B. cinerea* for 28 d was measured. Each treatment was replicated five times, and each experiment was conducted twice to confirm the results.(3)The body length, maximum body width, stylet length, esophageal gland length, gonad length, and tail length of the male and female RWTN fed transgenic *B. cinerea* for 28 d were measured. According to the morphological characteristics measurement method, RWTNs were killed by gentle heating, fixed in 4% FG fixative (40% formaldehyde: glycerol: distilled water = 10: 1: 89), dehydrated using the glycerol–ethanol method, and mounted on permanent slides [44]. Observations and measurements were carried out with a Scope A1 microscope equipped with a digital camera (AxioCam MRm; Zeiss, Oberkochen, Germany) and specialized Zen 2012 software (Zeiss). Twenty females and 20 males were randomly selected and measured for each treatment.(4)The pathogenicity of RWTN in *A. thaliana* was determined according to a previously described method [26]. One hundred mixed-stage nematodes from the T*n*, GT, and WT treatments were separately inoculated on the leaves of *A. thaliana* (Col-0). The symptoms in *A. thaliana* caused by RWTN were tested, and the nematodes were extracted from *A. thaliana* 21 days after inoculation. The rating of symptom severity in *A. thaliana* caused by foliar nematodes was assigned as follows: 0, no lesion/chlorosis; 1, 10% lesion/chlorosis; 2, 11–25% lesion/chlorosis; 3, 26–50% lesion/chlorosis; 4, 51–75% lesion/chlorosis; and 5, 75% or more lesion/chlorosis [26]. The blank control (CK) was treated identically, except that sterile distilled water was used. Each treatment was replicated five times, and each experiment was conducted twice to confirm the results.(5)The fat storage of RWTN were assessed using the Oil Red O staining method [45,46]. Images were captured using a Nikon Eclipse 90i microscope equipped with a digitizing tablet (NIS-Elements BR 4.10) and a digital camera (DS-RiI; Nikon, Minato City, Japan). Images were inverted to give a dark background and then auto-thresholded to identify the regions corresponding to nematodes. Within these regions, the levels of Oil Red O were quantified from the original images by determining the excess intensity in the red channel in comparison with the blue and green channels; regions with less red than blue or green were ignored. The mean fatness per image was estimated as the total intensity within the stained regions normalized by the area of the nematode regions [45]. Twenty adult nematodes were randomly selected and tested for each treatment.

### 4.7. Data Analyses

One-way ANOVA was performed using GraphPad Prism 8.2.1, and multiple comparisons were performed using Tukey’s test, with a significance level of *p* < 0.05. The *t*-test method was used to compare the experimental group and control group to determine if their mean values were significantly different (* *p* < 0.05, ** *p* < 0.01).

## Figures and Tables

**Figure 1 ijms-22-10057-f001:**
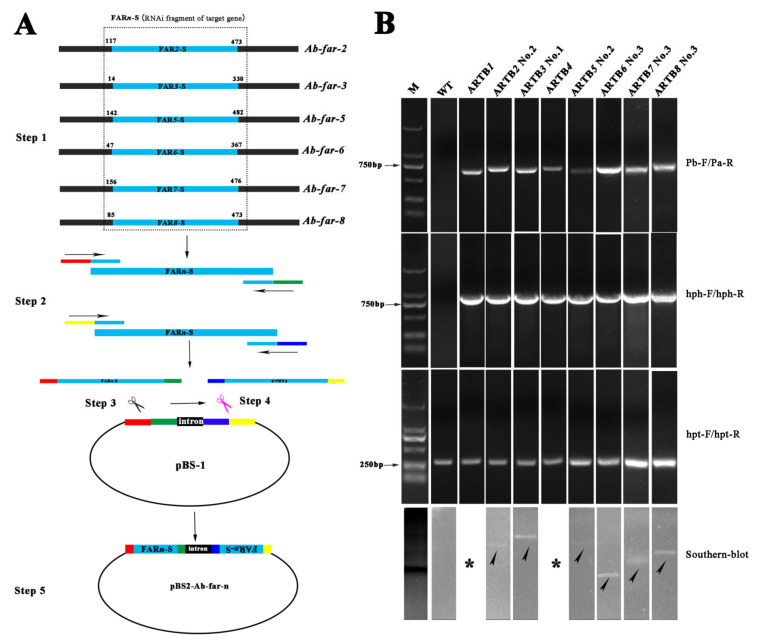
Production and identification of *Aphelenchoides besseyi Ab-far-n* hpRNA transgenic *Botrytis cinerea*. (**A**) Schematic of RNAi vector construction. Step 1: Selection of RNAi fragments, Step 2: PCR amplification of RNAi fragments, Step 3: Sense fragment insertion process, Step 4: Antisense fragment insertion process, and Step 5: Generation of final vector pBS2-Ab-far-n. (**B**) PCR identification and Southern blot of *Ab-far-n* hpRNA transgenic *B. cinerea*. M, marker DL2000; WT, wild-type *B. cinerea*; ARTB1-ARTB8, *Ab-far-1* to *Ab-far-8* hpRNA transgenic *B. cinerea*; No.1- No.3, strain 1–3. * Please refer the citation of the paper Ding et al. (2020).

**Figure 2 ijms-22-10057-f002:**
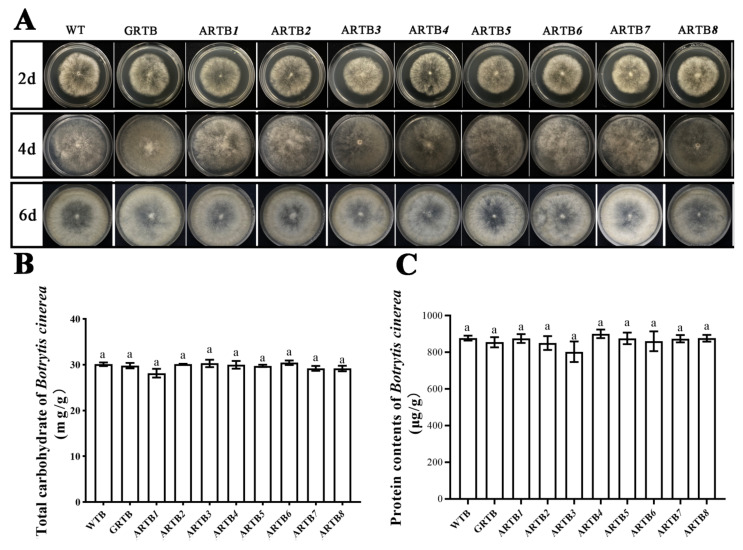
Growth, carbohydrate content, and protein content of *Ab-far-n* hpRNA transgenic *Botrytis cinerea*. (**A**) Growth of transgenic *B. cinerea*. (**B**) Total carbohydrate contents in transgenic *B. cinerea*. (**C**) Protein contents in transgenic *B. cinerea*. WTB, wild-type *B. cinerea*; GRTB, eGFP hpRNA transgenic *B. cinerea*; ARTB*1*-ARTB*8*, *Ab-far-1*—*Ab-far-8* hpRNA transgenic *B. cinerea*. Values are means ± standard error, n = 6. Same lowercase letters denote values that are no significantly different from each other (*p* < 0.05; Tukey’s test).

**Figure 3 ijms-22-10057-f003:**
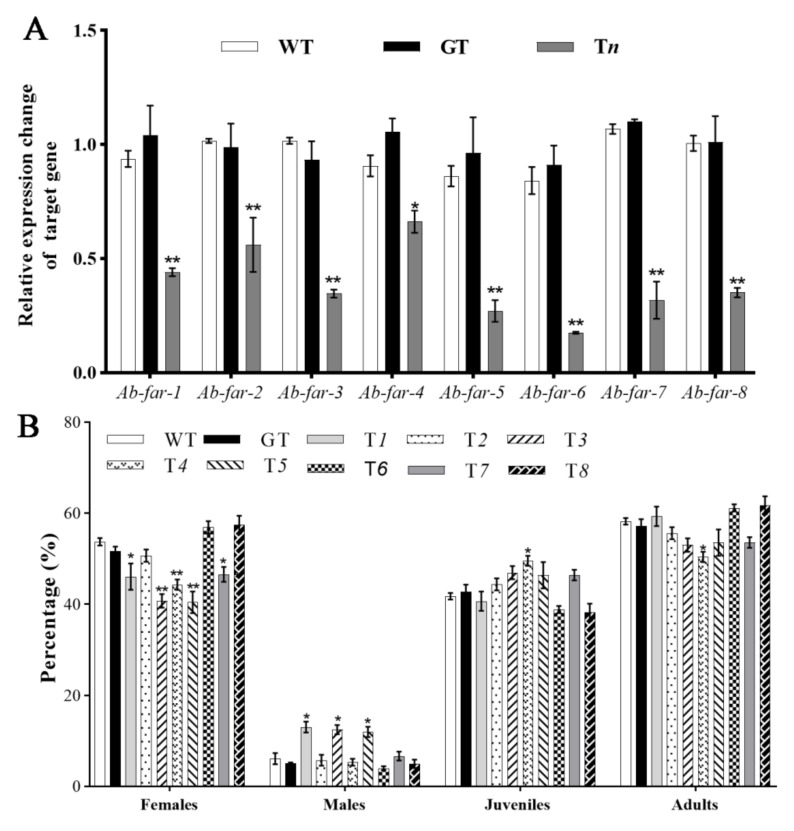
*Ab-far-n* expression and proportion of females, males, juveniles and adults of *Aphelenchoides besseyi* fed *Ab-far-n* hpRNA transgenic *Botrytis cinerea*. (**A**) *Ab-far-n* expression levels. (**B**) Proportion of females, males, juveniles and adults of *Aphelenchoides besseyi*. WT, nematodes cultured on wild-type *B. cinerea*; GT, nematodes cultured on eGFP hpRNA transgenic *B. cinerea*; T*1*-T*8*, nematodes cultured on *Ab-far-1* to *Ab-far-8* hpRNA transgenic *B. cinerea*. To determine silencing efficiency of gene, the gene expression from WT was normalized as 1 and fold change was calculated using the 2^-ΔΔCt^ method to determine change in expression of the genes of interest. Values of figure A are means ± standard error, n = 9. Values of figure B are means ± standard error, n = 10. Asterisks indicate values that are significantly different from the WT treatment (* *p* < 0.05, ** *p* < 0.01; *t*-test).

**Figure 4 ijms-22-10057-f004:**
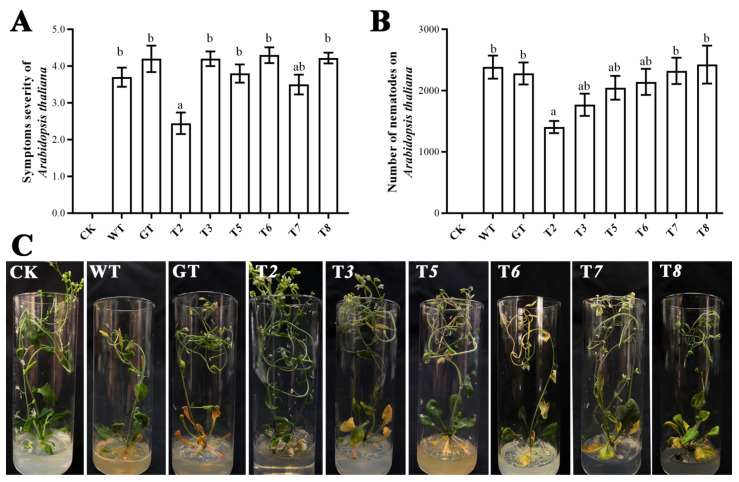
Pathogenicity of nematodes to *Arabidopsis thaliana*
*after* silencing *Ab-far-n* of *Aphelenchoides besseyi*. (**A**) Symptom severity of *Arabidopsis thaliana* inoculated with *A. besseyi*. (**B**) Reproduction of *A. besseyi* inoculated on *A. thaliana*. (**C**) Symptoms of *A. thaliana* caused by *A. besseyi*. CK, blank control; WT, nematodes cultured on wild-type *B. cinerea*; GT, nematodes cultured on eGFP hpRNA transgenic *B. cinerea*; T*2*/T*3*/T*5*-T*8*, nematodes cultured on either *Ab-far-2*, *Ab-far-3*, or *Ab-far-5*—*Ab-far-8* hpRNA transgenic *B. cinerea*, respectively. Values are means ± standard error, n = 10. Different lowercase letters denote values that are significantly different from each other (*p* < 0.05; Tukey’s test).

**Figure 5 ijms-22-10057-f005:**
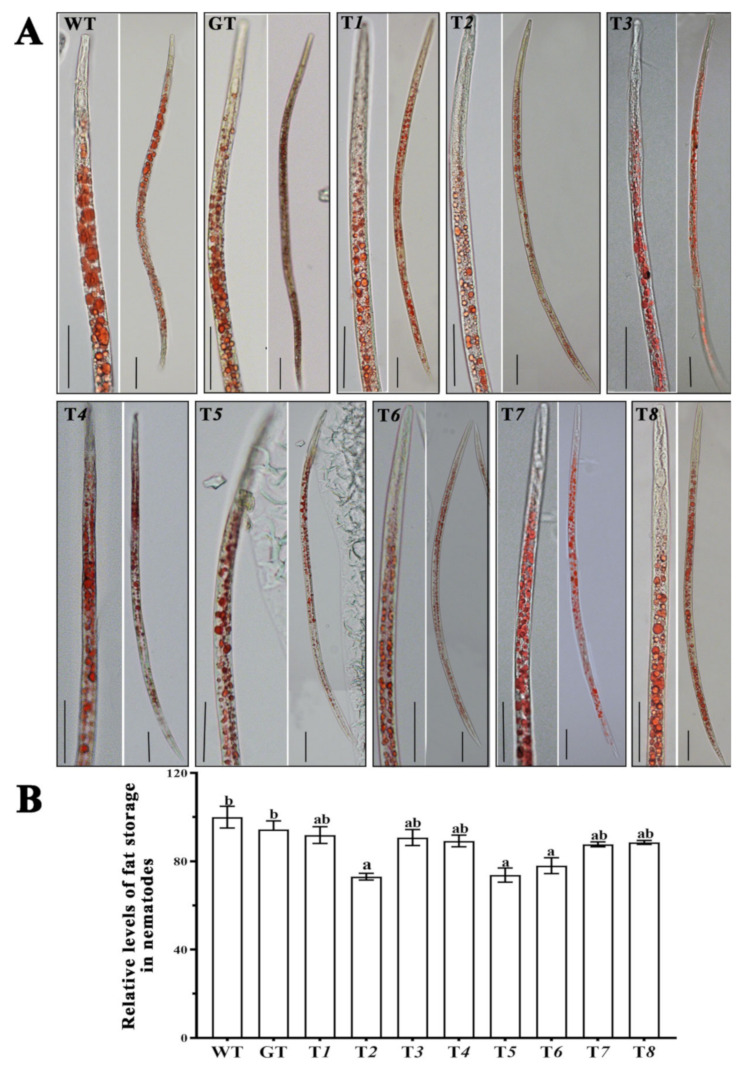
Fat storage of *Aphelenchoides besseyi* fed *Ab-far-n* hpRNA transgenic *Botrytis cinerea*. (**A**) Oil red O staining of *A. besseyi* cultured with transgenic *B. cinerea*. (**B**) The relative levels of fat storage in *A. besseyi* cultured with transgenic *B. cinerea*. WT, nematodes cultured on wild-type *B. cinerea*; GT, Nematodes cultured on eGFP hpRNA transgenic *B. cinerea*; T*1*-T*8*, nematodes cultured on either *Ab-far-1*—*Ab-far-8* hpRNA transgenic *B. cinerea*, respectively. Bar = 50 μm. Values are means ± standard error, n = 10. Different lowercase letters denote values that are significantly different from each other (*p* < 0.05; Tukey’s test).

**Table 1 ijms-22-10057-t001:** Identification of *Ab-far-n* hpRNA transgenic *Botrytis cinerea* (ARTBn) used in this study.

Codes for *Botrytis cinerea*	Number of ARTB*n* Strains	ARTB*n* Strain Used	PCR Product Size			Number of Copies
			Pb-F/Pa-R	Hpt-F/Hpt-R	hph-F/hph-R	
WT				271bp		
ARTB*1*	6 *	No.2 *	647bp	271bp	858bp	Single *
ARTB*2*	3	No.2	749bp	271bp	858bp	Single
ARTB*3*	4	No.1	709bp	271bp	858bp	Single
ARTB*4*	19 *	No.2 *	787bp	271bp	858bp	Single *
ARTB*5*	3	No.2	733bp	271bp	858bp	Single
ARTB*6*	2	No.3	713bp	271bp	858bp	Single
ARTB*7*	5	No.2	713bp	271bp	858bp	Single
ARTB*8*	3	No.1	781bp	271bp	858bp	Single

Note: * Please refer the citation of the paper Ding et al. (2020).

**Table 2 ijms-22-10057-t002:** *Ab-far-n* silencing efficiency and reproduction of *Aphelenchoides besseyi* fed *Ab-far-n* hpRNA transgenic *Botrytis cinerea* for 28 days.

Codes for Silencing Treatments	*Ab-far-n* hpRNA Transgenic *Botrytis cinerea*	Silencing Efficiency	Reproduction
WT			8647 ± 479.06 (cd)
GT			9182 ± 312.65 (cd)
T*1*	*Ab-far-1*	52.98%	4762 ± 253.99 (a)
T*2*	*Ab-far-2*	44.16%	2955 ± 458.35 (a)
T*3*	*Ab-far-3*	65.87%	5880 ± 528.32 (b)
T*4*	*Ab-far-4*	26.71%	5728 ± 457.71 (b)
T*5*	*Ab-far-5*	68.96%	8352 ± 447.85 (bcd)
T*6*	*Ab-far-6*	78.30%	6734 ± 391.39 (bc)
T*7*	*Ab-far-7*	69.38%	9134 ± 410.24 (cd)
T*8*	*Ab-far-8*	64.95%	10937 ± 877.25 (d)

**Notes:** WT, nematodes cultured on wild-type *B. cinerea*; GT, nematodes cultured on eGFP hpRNA transgenic *B. cinerea*; T*1*-T*8*, nematodes cultured on *Ab-far-1* to *Ab-far-8* hpRNA transgenic *B. cinerea*. Values are means ± standard error, n = 10. Different lowercase letters denote values that are significantly different from each other (*p* < 0.05, Tukey’s test).

**Table 3 ijms-22-10057-t003:** Morphological characteristics of females in *Aphelenchoides besseyi* fed *Ab-far-n* hpRNA transgenic *Botrytis cinerea*.

Characteristics	WT	GT	T*1*	T*2*	T*3*	T*4*	T*5*	T*6*	T*7*	T*8*
n	20	20	20	20	20	20	20	20	20	20
L	743.5 ± 24.3	738.6 ± 39.2	690.2 ± 21.3 *	699.6 ± 52.6 *	711.6 ± 24.7 *	667.5 ± 55.6 *	733.4 ± 23.4 *	667.2 ± 117.9 *	722.9 ± 37 *	716.2 ± 40.2 *
	(696.4–779.3)	(656.6–783.5)	(661.3–744.8)	(620.4–797.2)	(683.1–764.3)	(592.8–757.9)	(697.8–778.2)	(190.9–757.9)	(663.1–815.7)	(655.6–779.2)
M	15.4 ± 1.5	15.9 ± 1.4	14 ± 1.5	16.5 ± 1.7	16.8 ± 1.7	17.3 ± 1.6	14.4 ± 1.7	16.3 ± 1.3	15.4 ± 1.9	16.8 ± 1.7
	(11.4–17.4)	(13.5–19.1)	(11–16.4)	(13.9–20)	(14.2–19.4)	(14.7–19.4)	(11.4–17.1)	(13.6–18.7)	(10.3–18.9)	(13.5–21.3)
S	11.5 ± 0.5	11.1 ± 0.5	11.1 ± 0.6	16.3 ± 5.4	10.9 ± 0.9	10.7 ± 0.7	10.9 ± 0.9	11.1 ± 0.7	11 ± 0.9	10.9 ± 0.9
	(10.2–12.6)	(10.2–12)	(10.1–12)	(9.4–25.5)	(9.5–12.4)	(9.5–11.9)	(9.7–12.4)	(10–12.6)	(9–12.2)	(9.2–12.4)
E	69.4 ± 5.8	71.1 ± 3.5	70.6 ± 10.3	67.7 ± 15.8	66.7 ± 10.6	60.9 ± 13	71.7 ± 6.3	59.7 ± 12.4	65.5 ± 12.6	69.5 ± 6
	(59.5–81.7)	(61.8–75.9)	(58.1–97.4)	(39.2–95.5)	(52.8–85.3)	(41–83.9)	(62.8–81.6)	(38.1–80.8)	(28.9–81)	(59.6–80.7)
G	196.1 ± 25.3	199.6 ± 22	197.6 ± 13.8	185.4 ± 21.9 *	187.9 ± 25.8	177.5 ± 25.9 ***	196.1 ± 10.7	185.9 ± 27.9	198.2 ± 21.5	205.4 ± 22
	(161.6–240.6)	(150.7–242)	(180.1–221)	(150.6–214.8)	(125.2–225.9)	(110.8–218.9)	(180.3–216)	(124.7–217.6)	(158–235.1)	(171–237.2)
T	45.6 ± 2.5	43 ± 3.7	41 ± 2.4	40.5 ± 6.6	41.4 ± 3.4	39.7 ± 6.7	40.4 ± 5.5	41.7 ± 3.8	42.4 ± 4.8	41.2 ± 4.5
	(40–48.8)	(32.8–50.7)	(37.8–47)	(21.8–48.2)	(36.6–48.4)	(25.3–48.6)	(33–49.8)	(36.1–48.3)	(35.2–53.5)	(34.2–47.4)

**Notes:** WT, nematodes cultured on wild-type *B. cinerea*; GT, nematodes cultured on eGFP hpRNA transgenic *B. cinerea*; T*1*-T*8*, nematodes cultured on *Ab-far-1* to *Ab-far-8* hpRNA transgenic *B. cinerea*; *n*, Number of specimens observed; L, Body length; M, Maximum body width; S, Stylet length; E, esophageal gland length; G, Gonad length; T, Tail length. Data in the table are presented as the mean ± standard error, n = 20, and the number in the brackets represents the data range. Asterisks indicate values that were significantly different (*p* < 0.05, *t*-test) from the WT treatment.

**Table 4 ijms-22-10057-t004:** Phenotypic changes after silencing *Ab-far-n* of *Aphelenchoides besseyi*.

Target Genes	R	Sd	Bl	Gl	P	Fs	mL (Wang et al. 2018)
*Ab-far-1*	√	√	√		√		Body wall, gonad
*Ab-far-2*	√		√	√	√	√	gonad
*Ab-far-3*	√	√	√				Esophageal gland, gonad, nerve ring, intestine
*Ab-far-4*	√		√	√	√		Esophageal gland, gonad, body wall
*Ab-far-5*		√	√			√	Gonad, nerve ring, intestine, body wall
*Ab-far-6*			√			√	Esophageal gland
*Ab-far-7*			√				Gonad
*Ab-far-8*			√				Esophageal gland

**Notes:** “√” indicates the phenotype change. R, Reproduction; Sd, sex differentiation; Bl, body length; Gl, gonadal length; P, pathogenicity; Fs, fat storage; mL, mRNA localization.

## Data Availability

Data sharing is not applicable to this article as no new data were created or analyzed in this study. The ORFs and protein sequences of Ab-FAR-1 to Ab-FAR-8 were deposited in GenBank under the accession numbers AGA60308, KT387726 to KT387732.

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
