# Peer review of "Novel Functions of the Fatty Acid and Retinol Binding Protein (FAR) Gene Family Revealed by Fungus-Mediated RNAi in the Parasitic Nematode, Aphelenchoides besseyi"

_ijms, 2021, doi:10.3390/ijms221810057_

Round 1

Reviewer 1 Report

This manuscript present the functional study of 8 fatty acid and retinol genes (far) from Aphelenchoides besseyi using RNAi mediated by Botrytis cinerea. The manuscript follows the same structure and work presented by Ding et al. (2020), only presenting the other 6 genes Ab-far n (n=2, 3, 5, 6, 7, 8) since Ab-far-1 and Ab-far-4 were studied in Ding et al. 2020. The manuscript is carefully written, and the results understandable. 

Some corrections are needed before being considered for publication. 

Critical issues: 

- Although the authors make reference to Arabidopsis thaliana as suitable model for research of plant-nematode interactions (Wang et al., 2016), for plant pathologists, it is incorrect to consider Arabidopsis thaliana for pathogenicity tests since is not a natural host of A. besseyi and Koch’s postulate cannot be addressed. It is recommended to change the title of the section to avoid this issue and change the text accordingly.  

- Its is recommendable for the authors to indicate the time of maintenance of RNAi effects. In the 21 days in planta, the nematodes may loose the silencing effect. 

- If transgenic B. cinerea will be expressing hairpin RNA for nematode silencing, why was not RNA extracted from the transgenic fungus to confirm it? Extraction of genomic DNA will only indicate the presence of the fragment in the vector.  

Minor corrections

L5 - identification (by numbers) of authors affiliation. 

L13 - endoparasitic nematode

L41 - remove “some”

L43 - add the in “…proteins is the fatty acid…”

L52/L53 - Pp-far has also been studied by RNAi in planta (Vieira et al., 2017)

L60 - incorrect parasitic pathogenicity

L67 - remove organizational 

L96 - ID of fragments ARTBn different from the ID presented in Fig 1B (example No.2 ARTB2)

It seems that ARTB6 No.3 and ARTB5 No.5 present double bands, although faint but present. Is it possible to invert the image to confirm, like Ding et al., 2020. 

L101 - Please refer the citation of the paper Ding et al. (2020)

Fig.3 - Only females/males and juveniles are described in the legend. and the adults? Assuming that the authors are referring adult females and males. 

Table 1 and Table 2 with different codes for the treatments, even though Table 1 is for identification of transgenic B. cinerea and Table 2 for nematode silencing results. 

Fig. 5 - Italics in the first sentence of the legend. Correct Fat Stores to Fat storage (correct it also in the text). 

Images are very small. Zooming in loses quality. 

L221 - “….RWTN might also affect the fat storage…” since other possible rules are presented by the experiments conducted in this manuscript. 

L223 - Remove “fatty acid and retinol of”

Recommend Table 5 in the results and not in the Discussion. 

Esophageal or Oesophageal gland ? Please select one definition. 

Reviewer 2 Report

This reviewer has enjoyed very much the paper. My recommendation is to publish it in the present form. In fact, this is the first study to demonstrate that Ab-far genes affect sex formation and lipid metabolism in nematodes. Due the adventages for environment, the studies on the FAR gene family have become a hotspot in research for controlling Plant Parasitic Nematodes. This is a paper just in advanced science. The authors used a methodology in which they are very competent and authorities (RNAi, reverse genetics) as tool for study the function of these exclusive nematodes genes. The paper also reports as first time the specific functions of FAR proteins in Plan Parasitic Nematodes.

The paper is clearly written and results are evident regarding demonstration how gene silencing (RNAi) affects significantly to morphology, phenotype, pathogenesis and sex rate of Aphelenchoides beseyii. The conclusions derived through the discussion could be extended to most of the Plant Parasitic Nematodes.  

Author Response

Dear Editor and Reviewers,

 Thank you very much for your concerns and appreciates on our manuscript submitted. 

Best wishes,

Hui Xie